# Enhancing single-cell Multi-Modal Multi-Task Learning via Sparse Mixture-of-Experts

## Abstract

Recent advances in measuring high-dimensional modalities, including protein levels and DNA accessibility, at the single-cell level have prompted the need for frameworks capable of handling multi-omics data while simultaneously addressing multiple tasks. Despite these advancements, much of the work in the single-cell domain remains limited, often focusing on either a single-modal or single-task perspective. A few recent studies have ventured into multi-omics and multi-task learning, but we identified a ① **Optimization Conflict** issue, leading to suboptimal results when integrating additional modalities, which is undesirable. Furthermore, there is a ② **Costly Interpretability** challenge, as current approaches predominantly rely on costly post-hoc methods like SHAP. Motivated by these challenges, we introduce scMoE[1], a novel framework that, for the first time, applies Sparse Mixture-of-Experts (SMoE) within the single-cell domain. This is achieved by incorporating an SMoE layer into a transformer block with a cross-attention module. Thanks to its design, scMoE inherently possesses mechanistic interpretability, a critical aspect for understanding underlying mechanisms when handling biological data. Furthermore, from a post-hoc perspective, we enhance interpretability by extending the concept of activation vectors (CAVs). Extensive experiments on simulated dataset, Dyngen, and real-world multi-omics single-cell datasets, including {DBiT-seq, Patch-seq, ATAC-seq}, demonstrate the effectiveness of scMoE.

## 1 Introduction

Given the inherently multi-modal nature of multi-omics, which includes transcriptome, genome, and proteome data at the single-cell level (Lee et al., 2020), there exists a notable mismatch with current methodologies. These methods are predominantly tailored for single-modality applications, targeting specific tasks (Van Dijk et al., 2018; Yun et al., 2023; Xiong et al., 2019; Cheung et al., 2021), thereby limiting their generalizability in a multi-modal environment encompassing diverse tasks. Such tasks encompass the identification of joint groups, such as cell type across different modalities, and cross-modal prediction, where one modality is utilized to infer the expression of cells in another. Recently, UnitedNet (Tang et al., 2023) proposed a multi-task learning framework given its multi-modal nature, employing an encoder-common fuser-decoder framework based on a shared latent space. However, this approach encounters two fundamental limitations:

① **Optimization Conflict across Cell-Types and Multi-omics.** Compared to modalities we frequently observe in ML, modality conflict in multi-omics has not been thoroughly explored. As illustrated in Figure 1 (a), despite UnitedNet's capability in handling a diverse multi-modal environment, its peak performance is achieved using a subset of modalities (specifically, pre-MRNA and mRNA), rather than all four modalities, which paradoxically show the worst performance among the variations. This counterintuitive outcome, given the amount of information involved, is undesirable and highlights a limitation in harnessing the full potential of multi-omics data in the single-cell domain. A deeper investigation, as depicted in Figure 1 (b), reveals that the core issue stems from the encoder-common fuser-decoder framework in multi-task settings, leading to an optimization conflict. This conflict arises because the fuser consolidates information from each modality into a shared parameter space responsible for handling different tasks. This observation underscores the

---

[1] single-cell Sparse Mixture-of-Experts. Source code can be found in Supplementary files.

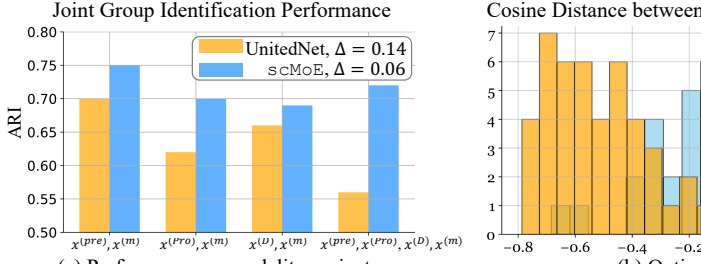

Figure 1: (a) Joint group identification performance across modality variants reveals that UnitedNet, when utilizing all modalities (Protein ($Pro$), mRNA ($m$), pre-mRNA ($pre$), DNA ($D$)), performs the worst. This is evidenced by a significant gap ($\Delta = 0.14$) compared to our proposed model ($\Delta = 0.06$), which maintains stable performance across a diverse range of modality combinations. (b) This phenomenon is attributed to optimization conflict issues, specifically gradient conflicts among modalities, such as "Protein" and "DNA" modalities. Here, the gradients are obtained from the experts and dense MLP with the same configuration in scMoE and Dense Model, respectively. Unlike currently adopted Dense Models like UnitedNet, our proposed sparse model demonstrates reduced conflict, as evidenced by more positive cosine distances, thereby facilitating enhanced multi-omics integration. The Dyngen dataset is used for the experiment.

need for a framework designed to disentangle the parameter space, allowing for the coexistence of both common and specialized knowledge tailored for diverse tasks.

② **Costly Interpretability of SHAP.** Interpretability in the biomedical and bioinformatics domain is essential, particularly for the practical application of these fields in clinical settings (Han & Liu, 2021; Karim et al., 2023). For instance, in predicting patient responses to cancer treatments using machine learning models, clinicians might hesitate to trust a model's recommendations if they lack interpretability. A model that can elucidate the genetic markers or pathways influencing its predictions enables practitioners to make more informed, potentially life-saving decisions. Although UnitedNet provides some level of interpretability through post-hoc analysis with the SHapley Additive exPlanations (SHAP) algorithm (Lundberg & Lee, 2017), this method has its drawbacks. It comes with a high computational cost due to its need to evaluate all possible combinations of features, a complexity that increases exponentially with the number of features. This poses a significant challenge in the bio domain, where providing relevant explanations in a timely manner is crucial. This brings the necessity for mechanistic interpretability, which is inherently integrated into the model's architecture, offering immediate insights during inference. Additionally, a lightweight design for post-hoc analysis that aligns with the needs of the bio domain is also vital.

⋆ **Sparse MoE as a Solution.** SMoE (Shazeer et al., 2017a) is an advanced neural network architecture that stands out for its ability to process complex, high-dimensional data efficiently. It achieves this by dynamically selecting a subset of specialized models, i.e., experts, for each input, offering a tailored approach in terms of a data-driven approach. Here, targeting challenges of single-cell multi-omics data, we propose scMoE, which replaces the MLP layer in a transformer architecture with an SMoE layer, which can naturally address the limitations of previous work. Specifically, it addresses the ① Optimization Conflict by employing multiple experts, thereby naturally disentangling the parameter space. This disentanglement allows for the attainment of both shared and unique knowledge tailored for each task and modality. Regarding the challenge of ② Costly Interpretability, scMoE addresses this by incorporating a gating network, or router, which automatically activates specific experts. This mechanism significantly enhances the model's interpretability by immediately identifying which experts are most relevant for the task at hand during inference. Moreover, the integration of a cross-attention module before the SMoE layer, based on transformer architecture (Fedus et al., 2022), further enriches interpretability. This module adeptly captures the importance of feature combinations from different modalities, facilitating solving downstream tasks with improved efficiency and insights[2].

In summary, our contributions are three-fold:

---

[2]In this paper, without further specification, one modality corresponds to one single-omic, thus the multi-modalities equal to the multi-omics.

- For the first time, we adopt the SMoE in single-cell multi-omics multi-task learning to effectively tackle both optimization conflict and costly interpretability issues.

- To enhance interpretability efficiently, we investigate the use of concept-activation vectors (CAVs), which are particularly suitable for the single-cell domain.

- We demonstrate the effectiveness of `scMoE` across diverse multi-omics single-cell datasets. This includes the simulations dataset `Dyngen`, as well as real-world datasets such as {`DBiT-seq`, `Patch-seq`, `DLPFSC`, `ATAC-seq`}, in joint group identification and cross-modal prediction tasks.

## 2 RELATED WORK

**Multi-Modal Multi-Task learning in single-cell data.** Multi-modal learning (Makadia et al., 2008; Weston et al., 2011; Antol et al., 2015; Goyal et al., 2017; Ramesh et al., 2022; Saharia et al., 2022; Yang et al., 2016; Dai et al., 2022; Jaegle et al., 2021) and multi-task learning (Xue et al., 2007; Zamir et al., 2018; Hashimoto et al., 2017; Fan et al., 2022; Chen et al., 2023) have been subjects of extensive research over the years, with significant contributions from various fields. Such advancements inspired single-cell domain where uni-modal targeting signle-task, e.g., transcriptome with imputation task (Li & Li, 2018; Van Dijk et al., 2018; Wang et al., 2021; Yun et al., 2023) or clustering task (Tian et al., 2019; Lee et al., 2023), was predominant. For instance, MOFA (Argelaguet et al., 2020) disentangs variation in single-cell studies integrating different omics data types, like genomics and proteomics. totalVI (Gayoso et al., 2021), on the other hand, specifically integrates single-cell RNA sequencing data and protein abundance for a comprehensive cellular profile. WNN (Hao et al., 2021) combines single-cell RNA and protein data, creating a unified representation of cell states. Schema (Singh et al., 2021) integrates diverse single-cell omics data, including transcriptomics and electrophysiology, providing a holistic view of cellular function and state. Most recently, UnitedNet (Tang et al., 2023) has been introduced targeting multi-tasks like joint group identification and cross-modal prediction by utilizing a shared-latent space in a post-hoc explainable manner. However, as mentioned earlier, it encounters an optimization conflict issue. Involving more modalities can significantly degrade overall performance while also imposing a substantial burden on post-hoc interpretability.

**Sparse Mixture-of-Experts (SMoE).** SMoE (Shazeer et al., 2017a) evolves from the traditional Mixture-of-Experts (MoE) model (Jacobs et al., 1991; Jordan & Jacobs, 1994; Chen et al., 1999; Yuksel et al., 2012) by incorporating sparsity into its structure, optimizing computational efficiency and model performance. This innovation allows SMoE to selectively activate only the most relevant experts for a given task, reducing the overhead and improving scalability, particularly beneficial in handling complex, high-dimensional datasets across diverse applications. It has seen rising use across vision (Riquelme et al., 2021; Lou et al., 2021; Eigen et al., 2013; Ahmed et al., 2016; Gross et al., 2017; Wang et al., 2020; Yang et al., 2019; Abbas & Andreopoulos, 2020; Pavlitskaya et al., 2020) and language processing (Lepikhin et al., 2021; Kim et al., 2021; Zhou et al., 2022; Zhang et al., 2021; Zuo et al., 2022; Jiang et al., 2021) fields. Its ability to dynamically assign different parts of the network to specific tasks (Ma et al., 2018; Aoki et al., 2021; Hazimeh et al., 2021; Chen et al., 2023) or data modalities (Kudugunta et al., 2021; Mustafa et al., 2022) has been explored for various applications. Research has shown its effectiveness in scenarios ranging from classification tasks in digital number recognition (Hazimeh et al., 2021) and medical signal processing (Aoki et al., 2021). However, its potential for generalization in the bio domain, especially in the area of single-cell research characterized by its multi-modal nature, remains unexplored.

## 3 METHOD

### 3.1 PRELIMINARIES

**Joint Group Identification with Cross-Modal Prediction.** Given a multi-modal single-cell data, we aim to solve a multi-task problem. The first task, joint group identification, is to identify jointly expressed characteristics, a commonality across cells despite their differing modalities such as cell type, states, or tissue regions. From a classification perspective, both unsupervised and supervised approaches can be utilized simply by modifying the loss function. Simultaneously, we aim to address the cross-modal prediction task, which infers the information, i.e., expression of cells in one modality,

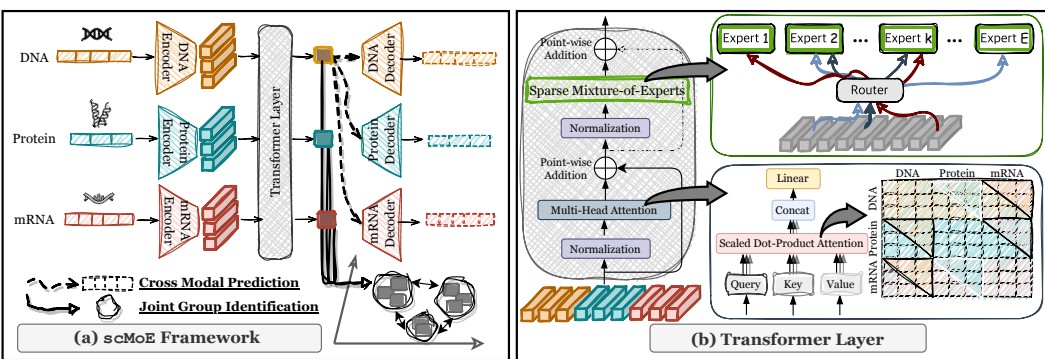

Figure 2: (a) The overview of scMoE: In the single-cell domain, each modality undergoes processing by its specific encoder, integrates with a shared transformer layer, and then passes through a modality-specific decoder. This structure enables simultaneous cross-modal prediction and group identification directly from the transformer output. (b) The transformer layer employs multi-head attention on concatenated modalities' inputs, facilitating intra-modality self-attention and inter-modality cross-attention. A Sparse Mixture-of-Experts Layer then supersedes the standard feedforward network, enhancing the model's efficacy in single-cell multi-modal multi-task learning.

using data from other modalities. Considering technical noise or properties that are difficult to measure, such predictions have the potential to significantly impact the real-world single-cell domain.

**Sparse Mixture-of-Experts (SMoE).** To address the optimization conflict issue identified previously, we implement SMoE to separate the parameter space across different tasks and modalities. In our model architecture, which is based on the transformer block, we substitute the traditional feed-forward neural network (FNN) with an SMoE layer, as depicted in Figure 2(b). Formally, the SMoE comprises several experts, denoted as $f_1, f_2, \ldots, f_E$, where E represents the total number of experts, and a routing mechanism, $\mathcal{R}$, which selects experts in a sparse fashion. For a given embedding $\mathbf{x}$, the top-$k$ experts are engaged by $\mathcal{R}$ based on the highest scores $\mathcal{R}(\mathbf{x})_i$, where $i$ indicates the expert index. This procedure is articulated as follows:

$$
\mathbf{y} = \sum_{i=1}^{k} \mathcal{R}(\mathbf{x})_i \cdot f_i(\mathbf{x}),
$$
$$
\mathcal{R}(\mathbf{x}) = \text{Top-K}(\text{softmax}(g(\mathbf{x})), k), \qquad (1)
$$
$$
\text{TopK}(\mathbf{v}, k) = \begin{cases} \mathbf{v}, & \text{if } \mathbf{v} \text{ is in the top } k, \\ 0, & \text{otherwise.} \end{cases}
$$

where $\mathbf{y}$, the final output of the SMoE layer, is a weighted sum of the expert representations $f_i(\mathbf{x})$ and their corresponding weights $\mathcal{R}(\mathbf{x})_i$, as determined by the router $\mathcal{R}$. Here, $g$ denotes a trainable network, typically a small FNN that ranges from one to a few layers (Shazeer et al., 2017b; Riquelme et al., 2021). The Top-K($\cdot$) operation selectively retains a vector $v$ if its probability is among the top $K$ probabilities; otherwise, it sets the vector to zero.

### 3.2 scMoE: single-cell meets SMoE

In essence, as illustrated in Figure 2, scMoE adopts the **Encoder-Transformer-Decoder** framework. Below, we detail each module accordingly.

**Encoder.** Given the multimodal nature originating from diverse environments, such as multi-omics, we initially employ modality-specific encoders, denoted as $\mathcal{E}^{(1)}, \cdots, \mathcal{E}^{(\mathcal{V})}$, where $\mathcal{V}$ represents the total number of modalities, to effectively generate informative embeddings for each modality. Notably, our use of transformer blocks (Fedus et al., 2022; Hu & Singh, 2021) requires input tokenization, differing from our matrix-formatted single-cell domain inputs where rows represent cells and columns denote specific modalities (e.g., genes or proteins), some of which, like highly variable genes (HVG) for the gene modality, vary in number. To achieve a consistent embedding shape across different modalities through tokenization, we adopt the patching method widely used in

ViT-based works (Dosovitskiy et al., 2021). Thus, for an input $\mathbf{x}^{(\nu)} \in \mathbb{R}^{B \times |\nu|}$, with batch size $B$ and size of a specific modality $|\nu|$, the output after processing by $\mathcal{E}^{(\nu)}$ is the embedding $\mathbf{h}^{(\nu)} \in \mathbb{R}^{B \times P \times D}$, where $P$ and $D$ denote the desired number of patches and the hidden dimension, respectively. With these tokenized embeddings from each modality, we proceed to the transformer block, the core component of this work.

**Transformer.** The transformer block, depicted in Figure 2 (b), primarily functions as a feature extractor and includes two key components: (1) Multi-Head Attention module facilitates both intra-modal and inter-modal attention through the modality-wise concatenated tokenized embeddings, $\mathbf{h} \in \mathbb{R}^{B \times P\mathcal{V} \times D}$. This setup enables the capturing of similarities between queries and keys across all modality combinations, with a total of $\mathcal{V}^2$. Representing the similarities in a matrix, diagonal elements would indicate self-attention within a modality (e.g., protein-protein) while off-diagonal elements signify cross-attention between different modalities (e.g., protein-mRNA), fostering a more thorough understanding of modalities. (2) The Sparse Mixture-of-Experts (SMoE) plays a crucial role in addressing optimization conflicts, thereby enhancing multi-task, multimodal environments as illustrated in Figure 1. By replacing the conventional FNN layers, SMoE enables the training of multi-experts who share common knowledge within modalities while retaining specialized knowledge in specific modalities or tasks. This is particularly pertinent in complex environments like the single-cell domain, where efficiency and specificity are essential. The output embedding from the transformer layer serves as a primary input for the unsupervised clustering loss, Deep Divergence-based Clustering (DDC) (Kampffmeyer et al., 2019), a method proven to enhance clustering in unsupervised settings. Notably, the DDC loss can be substituted with Cross-Entropy (CE) loss for supervised applications.

**Decoder.** In the single-cell domain, which often encounters noisy inputs due to dropout events (Hicks et al., 2018) and batch effects (Shaham et al., 2017), and lacks explicit supervision signals such as cell types, attaching decoder losses is a strategy to reconstruct the originally given input matrix effectively. Building on the final embeddings from the transformer block, we incorporate a total of $\mathcal{V}$ decoders, $\mathcal{D}^{(1)}, \cdots, \mathcal{D}^{(\mathcal{V})}$, meaning there is a decoder for each modality, similar to our approach with the encoder. Given our focus on the cross-modal prediction task—predicting the expression of one modality from another—and considering that each modality serves as both input to itself and to other modalities, we aggregate a total of $\mathcal{V}^2$ reconstruction losses.

**Training Procedure.** Facing a multi-task learning scenario, we aggregate two primary losses: the DDC loss (or CE loss in supervised contexts) and the Reconstruction loss, addressing joint group identification and cross-modal prediction tasks, respectively. Unlike the iterative loss update strategy employed in UnitedNet (Tang et al., 2023), our method is straightforward, enhancing adaptability for future expansions to additional modalities. The comprehensive algorithm for training scMoE is detailed in Algorithm 1.

---

**Algorithm 1** The overall procedure of scMoE.

---

1: **Input:** Cell matrices, $\mathbf{x}^{(\nu)}$, Encoders, $\mathcal{E}^{(\nu)}$, Decoders $\mathcal{D}^{(\nu)}, \forall \nu \leq \mathcal{V}$, with Transformer Layer containing MHA and SMoE
2: **Output:** Joint Group Identification, Cross-Modal Prediction
3: /* Encoder */
4: **for** $\nu = 1, \cdots, \mathcal{V}$ **do**
5:    $\mathbf{h}^{(\nu)} \leftarrow \mathcal{E}^{(\nu)}(\mathbf{x}^{(\nu)})$
6: **end for**
7: $\mathbf{h} \leftarrow [\mathbf{h}^{(1)}||\cdots||\mathbf{h}^{(\mathcal{V})}]$
8: /* Transformer */
9: $\mathbf{h}' \leftarrow \text{MHA}(\text{Norm}(\mathbf{h})) + \text{Norm}(\mathbf{h})$
10: $\tilde{\mathbf{h}} \leftarrow \text{SMoE}(\text{Norm}(\mathbf{h}')) + \text{Norm}(\mathbf{h}')$              ▷ Equation (1)
11: $\mathcal{L}_{\text{DDC}} = \text{DDC}(\tilde{\mathbf{h}})$
12: /* Decoder */
13: **for** $\nu = 1, \cdots, \mathcal{V}$ **do**
14:    **for** $\mu = 1, \cdots, \mathcal{V}$ **do**
15:       $\widehat{\mathbf{x}^{(\nu)}} \leftarrow \mathcal{D}^{(\mu)}(\tilde{\mathbf{h}}^{(\mu)})$
16:       $\mathcal{L}_{\text{Recon}} \leftarrow \mathcal{L}_{\text{Recon}} + \text{Recon}(\mathbf{x}^{(\nu)}, \widehat{\mathbf{x}^{(\nu)}})$
17: **end for**
18: **end for**
19: $\mathcal{L}_{\text{Final}} \leftarrow \mathcal{L}_{\text{DDC}} + \mathcal{L}_{\text{Recon}}$

---

### 3.3 INTERPRETABILITY OF scMoE

In the field of biology, particularly in single-cell analysis, the interpretability of a proposed model is paramount. We explore the interpretability of scMoE from two perspectives: Mechanistic and Post-hoc.

**Mechanistic Interpretability.** Mechanistic interpretability (Wang et al., 2022; Kästner & Crook, 2023) involves understanding a model's internal mechanisms and how they contribute to its decisions, observable during inference without additional training. While the integration of the SMoE layer might seem to obscure the model's interpretability, the preceding multi-head attention mechanism, which captures both intra and inter-modality significance, maintains a level of interpretability. Furthermore, the SMoE layer's gating network, or router [3], which decides which experts to activate for a given modality or task, allows for mechanistic interpretability through its data-driven decision-making process. This aspect will be demonstrated in the subsequent experimental section.

**Post-hoc Interpretability.** The mechanistic approach, while valuable, may not always be immediately transparent, as decisions such as expert selection are based on learned patterns. To complement the model's complex inner workings without delving into intricate details, the post-hoc approach (Zhang & Zhu, 2018; Zou et al., 2023) provides insights at the input level. While SHAP (Lundberg & Lee, 2017) has been recently applied in single-cell multimodal analysis, its complexity prompts us to propose a more lightweight, directly applicable interpretability method based on Concept Activation Vectors (CAV) (Kim et al., 2018b), tailored for the single-cell domain. Further details will be presented in Section 4.5.

## 4 EXPERIMENTS

### 4.1 EXPERIMENTAL SETTINGS

**Datasets.** To evaluate our method, we conducted experiments on **four** different datasets, including one simulated dataset and three real-world datasets. The simulated dataset is the Dyngen dataset, which contains $500$ cells, each with DNA, pre-mRNA, mRNA, and protein modalities comprising $100$ dimensions of features. This dataset was generated using the *Dyngen* software (Cannoodt et al., 2021). For real-world data, we utilized the Patch-seq GABAergic neuron dataset (*i.e.*, PatchSeq dataset), which provides morphological (M), electrophysiological (E), and transcriptomic (T) features from GABAergic interneurons in the mouse visual cortex (Gouwens et al., 2020). After applying the quality control procedures from previous research (Gala et al., 2021), 3395 neurons were available for E-T analysis and $448$ for M-E-T analysis. The Multiome ATAC+gene expression BMMCs dataset (*i.e.*, ATAC-seq dataset) combines gene expression and genome-wide DNA accessibility data from $10$ donors across $4$ tissue sites (Luecken et al., 2021). The DBiT-seq embryo dataset (DBiT-seq dataset) includes 936 spots, with data on mRNA expression, protein expression, and niche mRNA. For more details about these datasets please refer to Appendix B.

**Compared Methods.** To demonstrate the superior performance of scMoE on the particularly challenging joint group identification task, we compare it with 5 state-of-the-art (SOTA) multi-modal integration methods: UnitedNet (Tang et al., 2023), the weighted Nearest Neighbor (WNN) (Hao et al., 2021), Schema (Singh et al., 2021), Multi-Omic Factor Analysis (MOFA) (Argelaguet et al., 2020), and totalVI (Gayoso et al., 2021). Additionally, we introduce an "Identification only" baseline for a more exhaustive comparison, which focuses solely on the identification aspect of the task without the complexity of integrating multiple modalities. Subsequently, we benchmark the cross-modal prediction performance of scMoE against three carefully selected baselines: UnitedNet, WNN, and the aforementioned "Identification only" method. We used the hyperparameter setting proposed in their paper, and for our best setting, please refer to Appendix C.

### 4.2 SIMULATION STUDY

As shown in Table 1, when targeting the unsupervised joint group identification task, our proposed method, scMoE, achieves the best scores across various combinations of modalities. Using solely

---

[3]In this paper, we adhere to a single router shared across modalities to improve interpretability when analyzing experts.

Table 1: Joint group identification task measured by ARI in simulated Dyngen dataset upon modality combinations. The performance is averaged upon 5-fold cross-validation sets.

| | Dyngen | | | |
| --- | --- | --- | --- | --- |
| | Modality Combiations | | | |
| | Pre, m | Pro, m | D, m | Pre, Pro, D, m |
| scMoE | **0.75** | **0.70** | **0.69** | **0.72** |
| UnitedNet | 0.70 | 0.62 | 0.66 | 0.56 |
| Idenfication only | 0.49 | 0.41 | 0.44 | 0.50 |
| WNN | 0.43 | 0.46 | 0.43 | 0.46 |
| Schema | 0.55 | 0.57 | 0.56 | 0.56 |
| MOFA | 0.02 | 0.05 | 0.66 | 0.05 |
| totalVI | 0.07 | 0.02 | 0.02 | - |

Table 2: Cross-modal prediction task measured by $R^2$ in simulated Dyngen dataset upon modality combinations. The performance is averaged upon 5-fold cross-validation sets.

| | Dyngen | | | |
| --- | --- | --- | --- | --- |
| | Modality Combiations | | | |
| | Pre, m | Pro, m | D, m | Pre, Pro, D, m |
| scMoE | **0.41** | **0.90** | **0.67** | **0.62** |
| UnitedNet | 0.39 | 0.60 | 0.52 | 0.47 |
| Idenfication only | 0.20 | 0.28 | 0.25 | 0.35 |
| WNN | 0.21 | 0.26 | 0.22 | 0.36 |

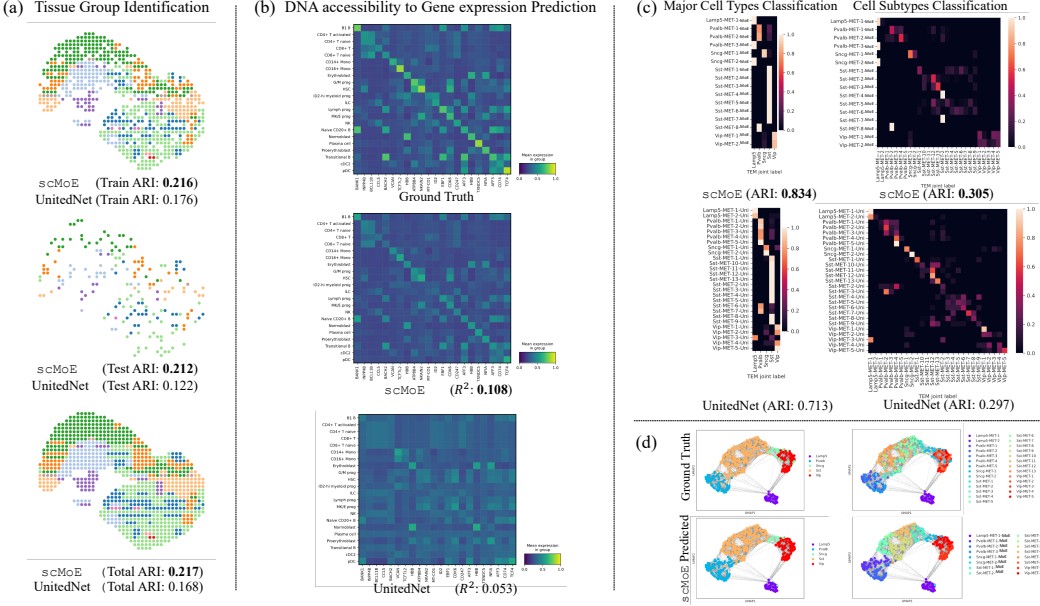

Figure 3: Performance analysis on real-world single-cell multimodal data: (a) Unsupervised tissue group identification task in the DBiT-Seq dataset. (b) Supervised cross-modal prediction task in the ATAC+gene expression BMMCs dataset. (c) Confusion matrix in Patch-seq dataset. (d) UMAP representation of latent features colored by joint cell types in Patch-seq dataset.

pre-mRNA, Protein, DNA, and mRNA is more beneficial than using two modalities, such as DNA and mRNA. This finding contrasts with the recently proposed UnitedNet, which tends to fall short when incorporating more modalities. As corroborated by the optimization conflict issue illustrated in Figure 1, scMoE benefits from a performance gain when adding more modalities. Furthermore, in Table 2, which focuses on the cross-modal prediction task, scMoE consistently outperforms the baselines, demonstrating its effectiveness and suitability for multi-modal and multi-task learning.

## 4.3 REAL-WORLD APPLICATIONS

In this section, we compare the scMoE with recent SOTA, i.e., UnitedNet, across various real-world datasets, including DBiT-seq, ATAC-seq, and Patch-seq. In Figure 3, we have following observations: **1)** In Figure 3 (a), we perform an unsupervised tissue group identification task to verify whether scMoE appropriately clusters 13 different groups. Compared to UnitedNet, our proposed model not only performs well in the training scenario but also in the unseen test scenario, verifying its generalizability. **2)** In Figure 3 (b), using the ATAC+gene expression BMMCs dataset, we

conducted a supervised cross-modal prediction task to infer gene expression given DNA accessibility. Sharing a common trend in the representation of each cell type (row), thanks to the supervised signal, scMoE exhibits a similar gene expression trend in each cell type compared to the Ground Truth, outperforming UnitedNet. **3)** Utilizing the Patch-seq dataset, we create a confusion matrix comparing the joint majority cell types and cell subtypes between reference labels and each model's identified label ('-MoE' for scMoE and '-Uni' for UnitedNet), as shown in Figure 3 (c). Unlike UnitedNet, where uncertain predictions hamper overall performance, scMoE shows a notable performance gain, especially in major cell-type classification. This improvement is attributed to the gating network in the SMoE layer, making the decision process both efficient and effective. This is further supported by the UMAP representation between Ground Truth and that of scMoE, showing its effectiveness in capturing relevant cell-type specific information between similar cells.

In conclusion, scMoE demonstrates significant generalizability across multiple multi-modal single-cell datasets, excelling in tasks such as unsupervised clustering and cross-modal prediction.

## 4.4 Ablation Study

To elucidate the contribution of the SMoE design within scMoE, we conduct a comprehensive series of ablation studies using the Dyngen dataset, specifically targeting the joint group identification task. We first examine the effects of SMoE by contrasting it with a dense model that possesses an equivalent number of parameters. Subsequently, our ablation analyses focus on the number of experts, the number of experts activated per token, the architecture of the routing network for SMoE, and the granularity of patches for each modality, aimed at understanding the impact on patch-level single-cell data representation. Within the Dyngen dataset, SMoE employs a single transformer block with the SMoE architec-

Table 3: Ablation Study of scMoE.

|  | Dyngen | | | |
|---|---|---|---|---|
|  | Modality Combiations | | | |
|  | Pre, m | Pro, m | D, m | Pre, Pro, D, m |
| scMoE | **0.97** | **0.83** | **0.89** | **0.75** |
| Dense | 0.68 | 0.68 | 0.71 | 0.72 |
| $N = 4$ | 0.82 | 0.71 | 0.77 | 0.75 |
| $N = 8$ | 0.80 | 0.69 | 0.76 | 0.75 |
| $N = 32$ | 0.74 | 0.65 | 0.76 | 0.74 |
| $k = 1$ | 0.79 | 0.68 | 0.73 | 0.70 |
| $k = 4$ | 0.82 | 0.70 | 0.79 | 0.74 |
| router per modality | 0.84 | 0.75 | 0.82 | 0.75 |
| 2 Patches per modality | 0.72 | 0.72 | 0.63 | 0.75 |
| 8 Patches per modality | 0.83 | 0.75 | 0.79 | 0.74 |
| 16 Patches per modality | 0.56 | 0.68 | 0.58 | 0.64 |

ture. This layer incorporates 16 experts, with 2 activated experts per token. We configure the model to process 4 patches per modality and utilize a single routing network for expert selection. The SMoE architecture demonstrates superior performance compared to its dense counterpart indicating the SMoE mitigates gradient conflicts among diverse modalities. Additionally, our findings reveal that both an excessively large and an insufficiently small number of experts can harmfully impact the model's performance. Similarly, the number of activated experts per token must be carefully calibrated, as too many or too few can also reduce model efficacy. We also observe that a modality-shared routing network outperforms a modality-specific routing network. Finally, our experiments identify that using 4 patches per modality represents a sweet point, with patch numbers beyond or below this threshold leading to diminished performance.

## 4.5 In-depth analysis of Interpretability

In this section, we demonstrate the interpretability of scMoE through two distinct approaches in the field of interpretable AI, i.e., mechanistic and post-hoc interpretability.

**Mechanistic Interpretability via SMoE and MHA.** A principal aspect of mechanistic interpretability is its capacity to provide plausible reasoning without necessitating additional training. We could observe two insightful interpretable aspects of scMoE as follows: **1)** In Figure 4 (a), we track on-the-fly expert selection process and found that different experts specialize in different modalities. For example, scMoE utilizes experts number $0, 5$, and $7$ to handle mRNA modalities and vice versa for DNA. On the other hand, scMoE framework can acquire both common knowledge, such as expert indices 3 and 4 adept at handling Pre-mRNA and Protein modalities. **2)** Moreover, an additional advantage of incorporating an attention module is observed through the consideration of both intra- and inter-modality relationships as shown in Figure 4 (b). Notably, significant cross-

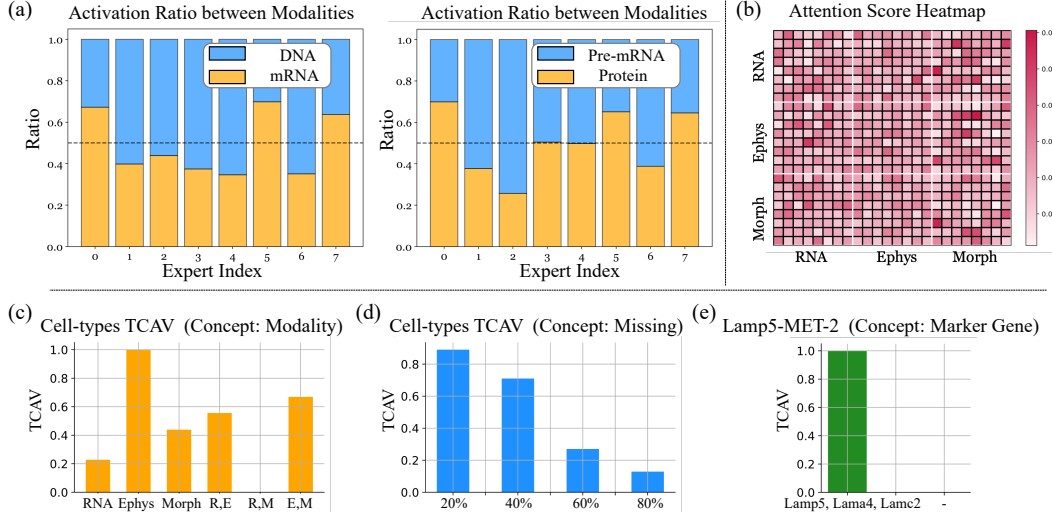

Figure 4: In-depth Analysis of Interpretability of scMoE. (a) Activation ratio between modalities in the Dyngen dataset. (b) Attention score heatmap in Patch-seq dataset. Post-hoc TCAV analysis on (c) how modality affects cell type classification, (d) how missing rate affects cell type classification, and (e) how marker genes affect rare cell type identification.

modality interactions, such as between Morph and Ephys, are evident in handling multi-modal data, as observed in the off-diagonal areas. Therefore, we argue that incorporating a Multi-Head Attention layer enhances mechanistic interpretability by underscoring the importance of considering inter-modal relationships in the multi-modal domain.

**Post-hoc Interpretability via TCAV.** Next, to further enhance interpretability in the single-cell domain, we propose a novel approach based on Testing with Concept Activation Vectors (TCAV) (Kim et al., 2018a). More specifically, our goal is to improve model interpretability through the identification of high-level concepts using sets of example data. To briefly introduce TCAV, a Concept Activation Vector (CAV) is generated by training a linear classifier to differentiate between examples of a concept and a random set of counterexamples. The CAV is then defined as the vector orthogonal to the classifier's decision boundary. In the analysis of specific instances, TCAV calculates directional derivatives to assess how the model's predictions are influenced by the core concept represented by the CAV. For instance, to evaluate the significance of the concept of stripes in an image of a zebra, a linear classifier is trained to distinguish between images of stripes and random samples. The CAV, orthogonal to the classification boundary, is utilized to gauge the sensitivity of the zebra image to the concept of stripes.

Inspired by TCAV, we propose a tailored approach for interpretability in the single-cell domain by designing concept vectors specifically suited for the biological domain. For example, leveraging the flexibility in defining concept vectors, we can identify which modality is crucial for a task or assess the impact of noisy and incomplete cell-gene matrix data (Figure 4 (c)), considering dropout events (Figure 4 (d)), or evaluate the efficacy of marker genes in classifying specific cell types (Figure 4 (e)). In Figure 4 (c), we observe that Ephys modality plays the most significant role in classifying cell types in Patch-seq dataset. Moreover, in scenarios of RNA missing, we observe that involving the genes with a lower missing rate (20%) is advantageous for downstream tasks as shown in Figure 4 (d). Finally, in identifying rare cell type, i.e., 'Lamp5-MET-2', which constituted only 2 samples out of 448, we define the concept with widely recognized marker genes such as Lamp5, Lama4, Lamc2. Surprisingly, selective sampling of cells highly expressing these marker genes revealed that classifying this rare cell type is feasible despite the limited sample size, highlighting the significant role of marker genes in the single-cell domain.

## 5 CONCLUSION

In this work, we investigate and design multi-modal multi-task learning algorithms targeting high-dimensional single-cell data through the lens of Sparse Mixture-of-Experts. Our pilot studies reveal two troublesome challenges in existing works, *i.e.*, *optimizaiton conflict* and *costly interpretability*. To fill in the missing research gap, we tailor the Sparse Mixture-of-Experts framework for single-cell data, which (1) disentangles the parameter space to allow modality- and task-specific modeling; (2) enjoys inherent mechanistic interpretability with enhanced post-hoc interpretability. Comprehensive validations on both simulated and real-world datasets consistently demonstrate the effectiveness of scMoE with enhanced interpretability. In the future, we are interested in extending our pipeline to systems immunology analysis with additional image modality. **Societal Impact:** Our work enhances biomedical research by improving model accuracy and interpretability, potentially leading to better healthcare outcomes and personalized medicine, but must address ethical considerations like patient data privacy and bias.

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

## A    APPENDIX

## B    DATASETS AND PREPROCESSING

**Dyngen Simulated Dataset.**    We use Dyngen (Cannoodt et al., 2021) to simulate a four-modality dataset comprising DNA, pre-mRNA, mRNA, and protein. The simulation generates 500 cells with each modality containing 100 dimensional features. Ground truth cell-type annotations are also provided. For Dyngen's parameters, we adopt the default settings of a linear backbone model as outlined in the Dyngen tutorial, employing functions such as `backbone_linear`, `initialize_model`, and `generate_dataset`.

**Patch-seq GABAergic Neuron Dataset.**    We utilize a Patch-seq dataset from GABAergic interneurons in the mouse visual cortex. It contains 3395 neurons retained for E-T analysis and 448 for M-E-T analysis. We standardize the input matrices for each modality to normalize the mean and standard deviation of all features in each cell to 0 and 1, respectively.

**Multiome ATAC + Gene Expression BMMCs Dataset.**    We analyze a multiome ATAC + gene expression dataset from BMMC tissue across 10 donors and 4 tissue sites. Following quality control and standard preprocessing procedures, we normalize the gene expression data using median normalization, log1p transform, and standardization. We select the top 4000 most variable genes via Scanpy. For DNA accessibility data, we binarize the matrix and select the top $13, 634$ most variable features, annotating DNA-accessibility peaks with ChIPseeker and `scanpy.var_names_make_unique`.

**UnitedNet on DBiT-seq Embryo Dataset.**    It includes three modalities: mRNA expression, protein expression, and niche mRNA expression from 936 spots. We normalize mRNA expression using `scanpy.pp.normalize_total` and select the top 568 differentially expressed genes. The protein expression is similarly normalized, focusing on 22 proteins, while niche modalities derive from normalized mRNA expression. For tissue region characterization, we extract ground truth labels from the original study and evaluate the clustering performance of UnitedNet against other methods using the adjusted rand index. For cross-modality prediction, we utilize mRNA and protein expression as inputs to UnitedNet. The DBiT-seq dataset is split into a training set ($80\%$, 748 spots) and a testing set ($20\%$, 188 spots) for the prediction task.

## C    HYPER-PARAMETER SETTING

**Dyngen.**    For the Dyngen dataset, we set the batch size to 64, the hidden dimension to 64, and train for 100 epochs. We represent each modality with 4 patches and use a learning rate of $1 \times 10^{-4}$. The model's architecture includes a single layer of transformer blocks with 1 attention head. In the Sparse Mixture-of-Experts (SMoE) component, we employ 16 experts, with 2 experts being activated simultaneously.

**DBiT-seq.**    For the DBiT-seq dataset, we configure the following parameters: a batch size of 64, a hidden dimension of 64, and 100 training epochs. Each modality is represented with 8 patches, and we employ a learning rate of $1 \times 10^{-4}$. Our model architecture consists of a single layer of transformer blocks with 4 attention heads. In the Sparse Mixture-of-Experts (SMoE) component, we utilize 8 experts, with 2 experts being activated at a time.

**Patch-seq.** The Patch-seq dataset experiments are conducted with a batch size of 32 and a hidden dimension of 128, over 100 training epochs. We represent each modality with 4 patches and set the learning rate to $4 \times 10^{-3}$. The model includes one layer of transformer blocks with 4 attention heads. The SMoE component comprises 32 experts, with 2 experts activated simultaneously.

**ATAC-seq.** In the ATAC-seq dataset, we use a batch size of 16 and a hidden dimension of 64, training for a total of 100 epochs. Modalities are represented with 8 patches each, and the learning rate is set to $1 \times 10^{-4}$. The architecture includes a single layer of transformer blocks with 1 attention head. We employ 8 experts in the SMoE component, with 2 experts being activated per token.

## D    IMPLEMENTATION DETAILS

All of our experiments were conducted using a single RTX3090 graphics card.

