# OpenReview forum: "Enhancing single-cell Multi-Modal Multi-Task Learning via Sparse Mixture-of-Experts"
_ICLR.cc/2025/Conference — ICLR 2025 Conference Withdrawn Submission_

### Official Review · Reviewer_SLMx · 2024-10-16

**Soundness:** 1
**Presentation:** 1
**Contribution:** 1
**Rating:** 1
**Confidence:** 5

**Summary:**

This paper proposes a new method based on MoE design for single-cell multi-modal analysis. They claim that their methods reach SOTA in their downstream applications.

**Strengths:**

It seems that their design can contribute some simple tasks in single-cell multi-modal data analysis.

**Weaknesses:**

I have several questions regarding the novelty and benchmarking design of their paper, which precludes their manuscript from acceptance. If the authors can fully address my questions, I may consider improving the score.

1. I believe that there have been many methods using MoE design for multi-modal data analysis [1-3]. Could the authors explain why they do not consider these methods into benchmarking analysis? And what is the difference between these methods and your method? I am pretty confused about the novelty here.

2. I do not find the information of hyper-parameter tuning for your methods and baselines. How can you ensure that your comparison is based on the optimal setting of different methods? Also, I wonder how do you compare the dense methods with MoE design and what is the difference of their scale. If the dense model has less parameter size with the MoE design, is it a fair comparison?

3. There are several benchmarking studies for single-cell multi-omic integration or label transformation. According to their study, it is necessary to include more baselines, including GLUE [4] and Seurat [5] (which is different from the WNN mentioned in this setting, I believe that the given WNN setting is from mudata, not from Seurat. Seurat is more complicated).

4. The current metrics only cover the label matching performances, but not the distance of different clusters. Therefore, I think the authors should include NMI or ASW for the clustering comparison.

References:

[1] https://www.cell.com/cell-reports-methods/fulltext/S2667-2375(21)00123-5

[2] https://journals.plos.org/ploscompbiol/article?id=10.1371/journal.pcbi.1009086

[3] https://ieeexplore.ieee.org/abstract/document/9778475

[4] https://www.nature.com/articles/s41587-022-01284-4

[5] https://pubmed.ncbi.nlm.nih.gov/29608179/

**Questions:**

Please see the weakness above.

**Details Of Ethics Concerns:**

I think the single-cell data contain information of patients or donors. How the authors can ensure that such information will not be leaked?

---

### Official Review · Reviewer_RSDj · 2024-10-21

**Soundness:** 2
**Presentation:** 2
**Contribution:** 3
**Rating:** 5
**Confidence:** 3

**Summary:**

This work introduces a framework (called scMoE) designed to address the challenges of single-cell multi-omics data analysis using Sparse Mixture-of-Experts (SMoE). The authors highlight two primary issues with existing approaches: optimization conflicts when integrating multiple modalities and costly interpretability. scMoE aims to resolve these issues by incorporating an SMoE layer into a transformer block with a cross-attention module, inherently providing what the authors call "mechanistic interpretability". The framework is evaluated through extensive experiments on simulated and real-world single-cell datasets, demonstrating its effectiveness in joint group identification and cross-modal prediction tasks.

**Strengths:**

- The application of SMoE in the single-cell domain is novel and addresses gaps in current methodologies issues.

-   The authors conduct extensive experiments on both simulated and real-world datasets, showing the robustness and generalizability of scMoE across different multi-modal single-cell data.

- The use of a transformer block with a cross-attention module enhances the model's ability to capture both intra- and inter-modality relationships, improving the overall performance.

- The paper includes an ablation study that elucidates the contributions of various components of the framework, such as the number of experts and the granularity of patches.

**Weaknesses:**

- While the paper demonstrates the effectiveness of scMoE on datasets of moderate size, the scalability of the framework to larger datasets with higher dimensions remains to be further validated.

 - The framework is supposed to inherently provide mechanistic interpretability through its design, which the authors claim is crucial for understanding complex biological data. However, I feel the so-called mechanistic interpretability and the resulting expert selection process are not really interpretable, especially for a non-technical person like a clinician or a biologist. Here more examples could help, how this method enhances the interpretability.

- the proposed post-hoc interpretability method (based on Concept Activation Vectors) is briefly mentioned but not extensively validated or compared with other post-hoc interpretability techniques. What concept vectors were designed specifically for the biological domain, and how were they chosen? Here it would be good to clarify and elaborate. I think it would be good to add a section about Concept Vector Activaiton to the appendix, for completeness.

- batch effects are crucial in single cell analysis. How is batch effect addressed with scMOE?

**Questions:**

see in the weakness section above.

minor comments
Figure 3: too small and not clear, need more explanation. E.g. 3a, what do the three different plots show and 3d, what are the 4 different plots?

minor things:
some typos:
- signle-task --> single task

---

### Official Review · Reviewer_bZhB · 2024-10-27

**Soundness:** 1
**Presentation:** 1
**Contribution:** 1
**Rating:** 3
**Confidence:** 4

**Summary:**

Here the authors propose a transformer architecture using sparse mixture of experts for analyzing single-cell multi-omics data. The authors apply their method to a simulated dataset and four real-world datasets.

**Strengths:**

* **Novelty**: To my knowledge, this is the first application of the sparse MoE framework to single-cell data.

**Weaknesses:**

Unfortunately the presented manuscript has a significant number of major issues that prevent me from recommending acceptance at this time. Details of my major concerns are below:

* **Missing experimental details:** The writing in the manuscript appears rushed, and is missing many experimental details. For example, what exact task corresponds to the results presented in Table 1? The caption mentions ARI, but the text never specifies what the ARI is being computed with respect to (e.g. cell type?). Are these labels obtained from the dyngen simulator? As another example, the authors mention an "Identification only" baseline method, but never specify any further details as to what this method entails.
* **Hyperparameter tuning:** In Appendix C the authors provide hyperparameter details for their method, and it appears that different settings were used for each dataset for scMoE. How were these parameters chosen? Without further justification it's impossible for the reader to know if such parameters were picked in a principled manner or were cherry-picked. This is especially concerning as the authors mention default parameters were used for all baseline methods.
* **Baseline methods/implementations:** Many of the baseline methods (e.g. those presented in Table 1) are not designed for the tasks that the authors applied them to. For example, totalVI's probabilistic model assumes paired RNA and surface protein measurements. Thus, it's not clear to me how the authors e.g. applied totalVI with only RNA or only protein measurements, and many of these comparisons seem unfair. Similarly, I'm not sure how one would apply the weighted nearest neighbors (WNN) method with only a single modality. Such issues may partially explain the significantly degraded performance of baseline methods compared to the authors' proposed method. To illustrate the merits of their approach, the authors should benchmark against methods that are actually designed for the corresponding tasks under consideration (e.g. for RNA-only experiments scVI could be used).
* **Underwhelming interpretability results**: The authors claim that previous methods for analyzing single-cell multi-omics data suffer from a lack of interpretability, but it's not clear to me how scMoE is a significant improvement on this front. The authors don't present any significant new insights derived from the expert activations/multi-headed attention mechanisms beyond perhaps giving some insight into their model's behavior (e.g. expert X uses modalities Y and Z). Moreover, TCAV isn't specific to the authors' proposed method and could be applied to any baseline model. Finally, some of the baselines noted in this work (e.g. MOFA) are linear methods, and thus are inherently far more interpretable. The bottom line is: what **new** insights does interpreting scMoE provide that couldn't be obtained with previous workflows?

**Questions:**

See "Weaknesses"

---

### Official Review · Reviewer_sF9D · 2024-10-29

**Soundness:** 2
**Presentation:** 3
**Contribution:** 2
**Rating:** 5
**Confidence:** 3

**Summary:**

This paper introduces a novel framework that integrates the sparse Mixture-of-Experts (SMoE) architecture into the scaled dot-product attention mechanism, offering a new multi-task learning approach for single-cell multi-omics analysis. The SMoE architecture provides a natural solution to enhance model interpretability and mitigate optimization conflicts, both of which are critical challenges in multi-modal and/or multi-task learning. Specifically, scMoE consists of $\mathcal{V}$ encoders which map the multi-modal data into embeddings with consistent shape. These embeddings are then passed into the SMoE-infused transformer block to learn intra-modal and inter-modal relationships. Finally, $\mathcal{V}$ decoders are used to generate reconstructions based on the learned embeddings from the transformer block. The proposed model is evaluated using both simulation study and real-world experiments to demonstrate its performance on the two tasks mentioned in the paper (i.e., joint group identification and cross-modal prediction).

**Strengths:**

•	Multi-modal and multi-task learning is an important but challenging topic in many scientific fields such as biology, public health, and environmental science. The paper provides a plausible solution to tackle the challenges related to model interpretability, optimization conflicts, and the complex interactions between modalities and tasks.

•	The methodology is presented in good flow and is easy to understand.

•	The authors compare scMoE with a reasonable number of existing models and perform ablation study to explore the optimal choices of model architecture and hyperparameters.

**Weaknesses:**

•	The introduction of the term "concept-activation vectors" on Line 111 feels abrupt. To enhance readability, it would be helpful to provide a brief explanation or context for this term earlier in the introduction, rather than deferring it until Section 4.5.

•	In related work, the authors briefly talked about the differences between traditional MoE and sparse MoE. However, the rationale for favoring sparse MoE over traditional MoE, particularly in the context of biological or multi-omics data, remains unclear. A more detailed explanation here would be appreciated.

•	The definition and calculation procedure of the ARI presented in Table 1 are not clearly explained.

•	The authors argue about significant cross-modality interactions between Morph and Ephys, but this does not seem to be obvious from Figure 4(b). Specifically, the off-diagonal block on (row 3, col 2) does not have higher attention scores compared to other blocks.

•	The appendix appears to be not well formatted.

Minor comments:

•	A small typo on Line 127 – “disentangles”.

**Questions:**

•	What does the term “multi-task” mean in the problem context if there are only two tasks (joint group identification and cross-modal prediction)? Based on the presented framework, the model is either doing unsupervised learning or generating predictions for all $\mathcal{V}$ modalities instead of learning separate representations for each task.

•	According to the illustration in Figure 2(a), for joint group identification task, did you apply clustering techniques to the learned representations from the transformer block? If so, did you map those representations into lower dimensions before doing clustering?

•	Have you evaluated scMoE using a single modality or is this just the “identification only” benchmark?

•	Why does scMoE not achieve the best performance with four modalities? Could this be due to the SMoE-infused attention block learning spurious correlations between different modalities?

---

### Note · Authors · 2024-12-02

I have read and agree with the venue's withdrawal policy on behalf of myself and my co-authors.